# Dim and Small Target Tracking Using an Improved Particle Filter Based on Adaptive Feature Fusion

**Youhui Huo** [1,2], **Yaohong Chen** [1], **Hongbo Zhang** [3], **Haifeng Zhang** [1,*] **and Hao Wang** [1,*]

1    Xi'an Institute of Optics and Precision Mechanics, Chinese Academy of Sciences, Xi'an 710119, China
2    University of Chinese Academy of Sciences, Beijing 100049, China
3    China Astronaut Research and Training Center, Beijing 100094, China
*    Correspondence: zhanghf@opt.ac.cn (H.Z.); wanghao@opt.ac.cn (H.W.)

**Abstract:** Particle filters have been widely used in dim and small target tracking, which plays a significant role in navigation applications. However, their characteristics, such as difficulty of expressing features for dim and small targets and lack of particle diversity caused by resampling, lead to a considerable negative impact on tracking performance. In the present paper, we propose an improved resampling particle filter algorithm based on adaptive multi-feature fusion to address the drawbacks of particle filters for dim and small target tracking and improve the tracking performance. We first establish an observation model based on the adaptive fusion of the features of the weighted grayscale intensity, edge information, and wavelet transform. We then generate new particles based on residual resampling by combining the target position in the previous frame and the particles in the current frame with higher weights, with the tracking accuracy and particle diversity improving simultaneously. The experimental results demonstrate that our proposed method achieves a high tracking performance with a distance accuracy of 77.2% and a running speed of 106 fps, respectively, meaning that it will have a promising prospect in dim and small target tracking applications.

**Keywords:** dim and small target; target tracking; feature fusion; particle filter; resampling method

## 1. Introduction

Video processing technologies such as target tracking [1,2], target detection [3,4], and moving target segmentation [5,6] have made great progress, with related applications continuing to emerge. Among them, target tracking has been widely used in intelligent video surveillance, modern military, and intelligent visual navigation equipment. A set of conventional methods have been proposed to meet the application requirements, including particle filters (PF) [7], kernelized correlation filters (KCF) [8], efficient convolution operators (ECO) [9], and spatial-temporal regularized correlation filters (STRCF) [10]. Dim and small target tracking is an important branch of target tracking, which plays a key role in military, aviation, and aerospace fields such as image matching guidance and reconnaissance [2]. Dim and small target tracking possesses more stringent requirements for trackers because the targets occupy only a small number of pixels in the image (the target size is generally about 2 × 2), which means that there is a lack of feature information, such as shape and texture, and a sensitivity to noise [11].

Recently, scholars have proposed many tracking algorithms to meet the requirements of the dim and small target tracking, which can be divided into two main categories including correlation filters-based methods [12–14] and PF-based methods [15–19]. KCF has received the most attention in the methods based on correlation filters for dim and small target tracking, which is achieved by establishing a discriminator based on the correlation operator with a kernel function. Qian K. et al. proposed an anti-interference small target tracking algorithm [12], which combines KCF with a detection model (KCFD), making a robust tracking result on image sequences with complex backgrounds. Zhang L. et al.

proposed an infrared small target tracking algorithm consisting of a discriminator based on KCF and a predictor based on the least-square trajectory prediction [13], which made full use of the continuity and direction of the target motion and can robustly track the target with shot-term occlusions. Kou Z. et al proposed a method based on target spatial distribution with improved KCF for infrared small target tracking [14], which considers the importance of intensity features to infrared targets and different regions to calculate the gray distribution weighting function of the target, solving the problems of target occlusion and drift. However, the above three methods based on KCF cannot achieve a higher tracking performance on the image sequences with a fast-moving target. A considerable number of previous studies on dim and small target tracking focus on particle filtering [15], which expresses the various motion states that the target possibly owns by the posteriori probability distribution estimated by a group of weighted particles, and it is a favorable method for solving nonlinear and non-Gaussian problems, fitting for dim and small target tracking. Conventional particle filtering has the following drawbacks during the dim and small target tracking task. First, the extracted features cannot explicitly represent the target and cause a tracking drift. Second, the conventional resampling method typically leads to a lack of particle diversity during the resampling process, which causes tracking performance degradation. Consequently, a considerable number of methods have been proposed to address the above drawbacks of particle filters for dim and small target tracking. Fan X. et al. proposed a particle filter method for adaptive template updating, combining the neighborhood motion model and the grayscale probability graph [16], which enhances the ability of dim and small target tracking. Ji E. et al. improved the mean-shift based particle filter tracking algorithm according to multi-feature fusion for small target tracking [17], which fuses a high-frequency histogram, fractal, and the energy of an infrared small target to improve the accuracy of the tracking process. Wang Y. et al. integrated the genetic resampling method into the particle filter method for small moving target tracking [18], which avoids particle degeneracy and guarantees particle diversity. Tian M. et al. proposed a track-before-detect method based on the spring model firefly algorithm optimization particle filter (SFA-PF-TBD) for dim small target tracking [19], making the distribution of particles more reasonable. However, these methods based on PF have a higher computational cost and may be prone to fall into local optimal values.

In the present paper, we propose a tracking algorithm based on adaptive feature fusion and an improved resampling particle filter, which is used to solve the two problems mentioned above of a particle filter for dim and small target tracking, namely a lack of the target features and the particle diversity, while reducing the computational cost of the algorithm. We first perform adaptive fusion for three features (grayscale intensity, edge information, and wavelet transform) based on the similarity of the features between the candidate region and the target. We then solve the problem of a lack of particle diversity by improving the residual resampling algorithm and generating some new particles that are meaningful for subsequent frames. In comparison to the methods PF [7], KCFD [12], and SFA-PF-TBD [19], our proposed method achieves a reasonable and robust tracking performance for six different dim and small target image sequences. The highlights and contributions of the present paper are as follows:

- Our method adaptively fuses three kinds of features to express the dim and small targets more accurately, making it robustly track the targets in various complex scenes.
- Our improved resampling method addresses the lack of particle diversity with a lower computational complexity, which makes a good balance between the tracking performance and computational cost.

This paper is organized as follows: Section 2 illustrates the contextual information for the particle filter algorithm; Section 3 exhibits the details of our method; Section 4 presents the experimental results and comparisons to other methods; and, finally, Section 5 concludes the paper and outlines the future directions.

## 2. Background Information

A particle filter is a nonlinear filtering algorithm based on the Monte Carlo method for Bayesian estimations, which is used to solve the complex multi-integral calculation problem in Bayesian filtering. Additionally, the central aim concerning a particle filter is to estimate the probability density with a set of weighted random samples (that is, particles) and then to perform a weighted summation on these particles to obtain the minimum variance estimate.

The state and observation equations of the target tracking system are set as follows:

$$X_t = f(X_{t-1}) + Q_t, Y_t = h(X_t) + R_t \tag{1}$$

where $X_t$ and $Y_t$ represent the state and observation values of the system at time $t$, respectively; $f(\bullet)$ and $h(\bullet)$ represent the system state transition function and observation function, respectively; and $Q_t$ and $R_t$ represent the system process noise and observation noise, respectively.

Bayesian filtering is generally divided into two parts: the prediction and update steps. The prediction step estimates the probability density of the current moment according to the posterior probability density of the previous moment:

$$p(X_t|Y_{1:t-1}) = \int p(X_t|X_{t-1})p(X_{t-1}|Y_{1:t-1})dX_{t-1} \tag{2}$$

The update step updates the posterior probability density of the current moment according to the estimated probability density and observed value of the current moment:

$$p(X_t|Y_{1:t}) = \eta[p(Y_t|X_t)p(X_t|Y_{1:t-1})] \tag{3}$$

where $\eta = [\int p(Y_t|X_t)p(X_t|Y_{1:t-1})\,dX_t]^{-1}$. Then, calculate the expected value of the posterior probability $p(X_t|Y_{1:t})$ to obtain the final estimated state value:

$$\hat{X}_t = \int X_t p(X_t|Y_{1:t})\,dX_t \tag{4}$$

The particle filter samples the posterior probability density based on the Monte Carlo method. However, it is difficult to directly sample the posterior probability in the actual situation, so an importance function, $q(X_t|Y_{1:t})$, is introduced, and Equation (4) is changed to the following:

$$\hat{X}_t = \int X_t \frac{p(X_t|Y_{1:t})}{q(X_t|Y_{1:t})} q(X_t|Y_{1:t})dX_t \tag{5}$$

Then, we can apply the Monte Carlo method to sample $q(X_t|Y_{1:t})$ to obtain

$$\hat{X}_t \approx \sum_{i=1}^{N} w^i{}_t X^i_t \tag{6}$$

where the particle weight $w^i{}_k$ is defined as follows:

$$w^i{}_t = \frac{p(X^i_t|Y_{1:t})}{q(X^i_t|Y_{1:t})} \tag{7}$$

Transform $p(X_t|Y_{1:t})$ and $q(X_t|Y_{1:t})$ to

$$p(X_t|Y_{1:t}) \propto p(Y_t|X_t)p(X_t|X_{t-1})p(X_{t-1}|Y_{1:t-1}) \tag{8}$$

$$q(X_t|Y_{1:t}) = q(X_t|X_{t-1}, Y_{1:t})q(X_{t-1}|Y_{1:t-1}) \tag{9}$$

Based on Equations (7)–(9), we can obtain the recurrence equation of the weight $w^i{}_k$:

$$w^i{}_t \propto w^i{}_{t-1} \frac{p(Y_t|X^i{}_t)p(X^i{}_t|X^i{}_{t-1})}{q(X^i{}_t|X^i{}_{t-1}, Y_{1:t})} \tag{10}$$

Equations (6), (9) and (10) constitute the main framework of particle filtering. Firstly, the importance function of the current moment is calculated according to the importance function at time $t-1$ and Equation (9). Then, using Equation (10) allows us to obtain the particle weight $w^i{}_k$. Finally, according to Equation (6), the estimated value of the state in the current moment is obtained.

## 3. Principle of the Proposed Method

### 3.1. Feature Extraction

Although the gray histogram feature can clearly express the grayscale distribution of the image, it lacks the description of the spatial position distribution, so it can be considered to obtain the gray histogram feature by combining the weighted position information [20]. We introduced a kernel function $k(r)$:

$$k(r) = \begin{cases} 1 - r^2, r < 1 \\ 0, r \geq 1 \end{cases} \tag{11}$$

which assigns different $k$ values based on the distance $r$. The longer the distance $r$, the small the $k$ value. Then, the gray feature with the position distribution information $f_{gray}$ can be expressed as follows:

$$f_{gray} = \alpha \sum_{i=1}^{n} k(\frac{||x_0 - x_i||}{s}) \delta[b(x_i) - m] \tag{12}$$

where $n$ represents the number of pixels in the target area; $m$ is the gray value range $[0, 255]$; $x_0$ is the position coordinate of the center pixel; $x_i$ is the position coordinate of the $i$-th pixel; $b(x_i)$ is the gray level of the pixel $x_i$ in the obtained histogram grade value; $s = \sqrt{H_x{}^2 + H_y{}^2}$ ($H_x$ and $H_y$ are the half-width and half-height of the area rectangle); and $\alpha$ is the normalization coefficient defined as follows.

$$\alpha = [\sum_{i=1}^{n} k(\frac{||x_0 - x_i||}{s})]^{-1} \tag{13}$$

The image edge information also plays a significant role in the feature of the image, which is a collection of pixels where the distribution of the image features, such as pixel gray value and texture, are discontinuous or where there is a step change. Image edge features have a certain robustness towards illumination changes and rotations, which can make up for the defects of gray features. A high number of methods have been developed to calculate the edge information [21], which can express the target in which we are also interested. In this paper, we chose the Sobel operator to establish the edge histogram based on the edge gradient amplitude.

The Sobel operator performs pixel-by-pixel convolution of the image to be processed and the template in the form of a sliding window to obtain edge information. We choose the Sobel operator template in four directions (horizontal $S_x$, vertical $S_y$, 45 degrees $S_{45°}$, and 135 degrees $S_y$) to extract image edge information in all aspects. The four operator templates are as follows:

$$S_x = \begin{bmatrix} 1, & 2, & 1 \\ 0, & 0, & 0 \\ -1, & -2, & -1 \end{bmatrix}, S_y = \begin{bmatrix} 1, & 0, & -1 \\ 2, & 0, & -2 \\ 1, & 0, & -1 \end{bmatrix}, S_{45°} = \begin{bmatrix} 2, & 1, & 0 \\ 1, & 0, & -1 \\ 0, & -1, & -2 \end{bmatrix}, S_{135°} = \begin{bmatrix} 0, & 1, & 2 \\ -1, & 0, & 1 \\ -2, & 1, & 0 \end{bmatrix} \tag{14}$$

Convolve the four templates $S_x, S_y, S_{45°}, S_{135°}$ with the original image $I$ pixel by pixel to obtain the edge gradient values in four directions:

$$G_x = S_x * I, \ G_y = S_y * I, \ G_{45°} = S_{45°} * I, \ G_{135°} = S_{135°} * I \tag{15}$$

where $*$ represents pixel-wise convolution. We then calculate the total edge gradient magnitude $G$ by synthesizing the edge gradient values in the four directions:

$$G = \sqrt{G_x{}^2 + G_y{}^2 + G_{45°}{}^2 + G_{135°}{}^2} \tag{16}$$

Finally, the edge histogram $f_{edge}$ can be extracted based on the kernel function presented in Equations (11) and (12).

To better express the target features, we considered combining the frequency domain features of the image. We chose the wavelet features because the wavelet transform has good time–frequency localization characteristics, which is convenient for adjusting the filter direction and fundamental frequency bandwidth, so as to better take into account the resolution of the spatial and frequency domains [22]. Moreover, the wavelet feature is insensitive to illumination changes and can tolerate target rotation and deformation to a certain extent.

The two-dimensional wavelet function $H(x, y)$ can be expressed as follows:

$$H(x, y) = \frac{1}{2\pi\sigma_x\sigma_y} \exp(-\frac{1}{2}(\frac{x^2}{\sigma_x{}^2} + \frac{y^2}{\sigma_y{}^2})) \cos(\omega x) \tag{17}$$

where $\sigma_x$ and $\sigma_y$ are the standard deviations on the $x$ and $y$ axes, respectively, which determine the size of the filtering area. $\omega$ is the baseband bandwidth. By convolving the original image $I$ with $H(x, y)$, we can obtain the wavelet transform $W$ of $I$:

$$W(x, y) = H(x, y) * I(x, y) \tag{18}$$

Here, $*$ represents the convolution operation. Similarly, Equations (11) and (12) can be used to extract the histogram $f_{wavelet}$ of $W$.

### 3.2. Feature Fusion and Model Establishment

We set the fusion weight by comparing the correlation $c_i$ between the three features of the initial target $f\_t_i$ and the three features of the current target $f\_c_i$ to perform adaptive feature fusion, where $i = (gray, edge, wavelet)$. The correlation calculation formula is as follows:

$$c_i(f\_t_i, f\_c_i) = \frac{\sum\limits_{j}^{J} [f\_t_i(j) - \overline{f\_t_i}][f\_c_i(j) - \overline{f\_c_i}]}{\sqrt{\sum\limits_{j}^{J} [f\_t_i(j) - \overline{f\_t_i}]^2 \sum\limits_{j}^{J} [f\_c_i(j) - \overline{f\_c_i}]^2}} \tag{19}$$

where $J$ represents the total number of feature histograms, $\overline{f\_t_i}$ and $\overline{f\_c_i}$ are the mean of $f\_t_i$ and $f\_c_i$, respectively. We then obtained the correlation $c_i$ of two sets of gray features, two sets of edge features, and two sets of wavelet features ($c_{gray}$, $c_{edge}$, and $c_{wavelet}$, respectively) based on Equation (19), and converted the correlation to the weight according to Equation (20):

$$v_i = \frac{0.5(c_i + 1)}{\sum_i 0.5(c_i + 1)}, \ \ i = (gray, edge, wavelet) \tag{20}$$

Then, the adaptive fusion feature is expressed as follows:

$$f = \sum_i v_i f_i, \ \ i = (gray, edge, wavelet) \tag{21}$$

In the process of particle filter target tracking, it is necessary to update the estimate by using the observation value in the current moment. In the present paper, we chose to update the similarity between the fusion histogram feature $f_p$ of the target model and the fusion histogram feature $f_q$ of the candidate region. The similarity measure uses the Bhattacharyya distance $d$:

$$d = \sqrt{1 - \rho(f_p, f_q)} \tag{22}$$

where $\rho(f_p, f_q) = \sum_{i}^{M} \sqrt{f_p^{(i)} f_q^{(i)}}$ and $M$ is the histogram dimension. The larger $\rho$ is, the more similar the candidate region histogram is to the target model histogram, that is, the candidate region is likely to be the estimated target position, and the particle will be given a higher weight.

### 3.3. Improved Resampling Particle Filter Algorithm

The resampling method generally adopted in the conventional particle filter algorithm is to simply copy the particles with high weights and to discard the particles with low weights, which will cause some particles with high weights to be sampled multiple times, resulting in a lack of samples and missing particle diversities [1]. In the current paper, based on residual resampling, we generated new particles according to high-weight particles in the current moment and the target position of the previous frame to solve the problem of a lack of particle samples, and the newly added particles can also track subsequent frames better.

First, we defined an effective particle number $N_{eff}$ to judge whether the particles were degenerated, which is calculated based on weights of all particles and can clearly reflect the weight distribution of the entire particle set. $N_{eff}$ is defined as follows:

$$N_{eff} = \left[\sum_{i=1}^{N} \left(w_k^i\right)^2\right]^{-1} \tag{23}$$

where $N$ is the total number of particles. The smaller the number of effective particles $N_{eff}$, the more uneven distribution of particle weights, that is, there may be a large number of particles with small weights and a few particles with large weights, which means that particle diversity is missing.

We set a threshold $N_{th}$ ($N_{th} = \frac{2}{3}N$ in this paper); if $N_{eff} \geq N_{th}$, we assumed that the particles had a certain diversity and no resampling was required; if $N_{eff} < N_{th}$, it was considered that the particles degenerated, and residual resampling needed to be performed. Residual resampling consists of two parts as follows:

(1). We copied the valid particles, and the number of copies was determined by the weight of the particles. For the particle set $[P_{k\_old}^{(i)}, w_{k\_old}^{(i)}]$, we preserved $n_k^{(i)} = \left\lfloor N \cdot w_{k\_old}^{(i)} \right\rfloor$ particles and then obtained set $[P_{k\_r}^{(i)}, w_{k\_r}^{(i)}]$, where $\lfloor \rfloor$ represents the rounding operation, the corrected weights $w_{k\_r}^{(i)} = (N \cdot w_{k\_old}^{(i)} - n_k^{(i)})/N'$, and $N' = \sum_{i=1}^{N} n_k^{(i)}$.

(2). When the number of residual particles $M = N - N'$ was greater than 0, we randomly resampled the particles $[P_{k\_r}^{(i)}, w_{k\_r}^{(i)}]$ and sorted them according to their weight to obtain the particle set $[P_k^{(i)}, w_k^{(i)}]$.

The positions close to the particles with higher weights are likely to be the potential position of the target in the subsequent frame, which may be the particles useful for solving the problem of a lack of particle diversity and for facilitating the tracking of the subsequent frame. Therefore, we chose the first $J$ particles $P_k^{(j)} = (x_k^{(j)}, y_k^{(j)})$, $(j = 1, \ldots, J)$ in $[P_k^{(i)}, w_k^{(i)}]$ with high weights and adopted the points with a distance $\Delta d$ ($\Delta d = 2$, in this

paper) around $P_k{}^{(j)} = (x_k{}^{(j)}, y_k{}^{(j)})$, $(j = 1, \ldots, J)$ as the new particle candidate positions, as follows:

$$P_{k\_new}{}^{(j)} = ([x_k{}^{(j)}; 1] \cdot \Delta x, [y_k{}^{(j)}; 1] \cdot \Delta y) \tag{24}$$

where $\Delta x$ and $\Delta y$ represent the coordinates of the points around a particle with a distance of $\Delta d$ in eight directions, and $\Delta x$, $\Delta y$ are defined as Equation (25). Therefore, according to Equation (24), the points around the particles with the larger weights in the particle set $P_k{}^{(j)} = (x_k{}^{(j)}, y_k{}^{(j)})$, $(j = 1, \ldots, J)$ can be extracted as the new candidate particles.

$$
\begin{aligned}
\Delta x &= \begin{bmatrix} 1 & 1 & 1 & 1 & 1 & 1 & 1 & 1 \\ \Delta d & -\Delta d & 0 & 0 & \Delta d & \Delta d & -\Delta d & -\Delta d \end{bmatrix} \\
\Delta y &= \begin{bmatrix} 1 & 1 & 1 & 1 & 1 & 1 & 1 & 1 \\ 0 & 0 & \Delta d & -\Delta d & -\Delta d & \Delta d & \Delta d & -\Delta d \end{bmatrix}
\end{aligned} \tag{25}
$$

We can obtain a total of $l$ candidate particles by eliminating the repeated values appearing in $P_{k\_new}{}^{(j)}$ and can calculate the distance $d_o$ between the positions of these $l$ candidate particles and the estimated target position at the last moment. The first $0.2N$ candidate particles with the closest distance were used to substitute the $0.2N$ particles with low weights in $P_k{}^{(i)}$, and a new particle set was obtained. This not only can generate new particles to solve the problem of a lack of diversity of the traditional resampling particles but also can gather more effective particles at the potential positions of the target in the subsequent frame to facilitate subsequent tracking.

*3.4. Algorithm Process*

The pseudo code of the algorithm in this paper is shown in Algorithm 1.

---

**Algorithm 1.** The pseudo code of the algorithm we proposed.

---

Input: image sequences $I$ and target position in initial frame.
Output: the target position $Pos$ in subsequent frames.
% Initial frame. Initialization
Extract $f_{gray}$, $f_{edge}$, $f_{wavelet}$; and fuse them to get $f_p$;
Initialize $N$ particles and perform importance sampling.
% Subsequent frames. Tracking
for frame=2: length($I$):
    Extract $f_{gray}, f_{edge}, f_{wavelet}$ at current frame;
    Fuse $f_{gray}, f_{edge}, f_{wavelet}$ to get $f_q$ according to Equations (19)–(21);
    $d \leftarrow$ Equation (22);
    $w^i{}_k \leftarrow$ Equation (10);
    $Pos \leftarrow$ Equation (6);
    % Resampling
    if $N_{eff} \geq N_{th}$:
        continue;
    else:
        Perform residual resampling;
        Generate new particles according to Equation (24).
    end if
end for

---

## 4. Experimental Results and Analysis

To evaluate the tracking performance of the proposed method relative to others, the results of the tracking experiments were compared to the methods of PF [7], KCFD [12], and SFA-PF-TBD [19]. The experimental data adopted the data set published by Hui B. et al. [23], in which six groups of sequences, data5, data8, data16, data18, data19, and data20, were chosen for the tracking experiment. The resolutions of the images of six sequences were $256 \times 256$, and more specific descriptions about the six sequences are shown in Table 1. For a fair comparison of the tracking performance and computational

cost, we performed experiments using the four methods by running them on a laptop with 2.40 GHz Intel i5–1135G7 and 16GB RAM, and with MATLAB 2018a.

**Table 1.** Detailed description of six groups of sequences.

| Sequence | Total Frames | Target Size | Sequence Characteristics |
| --- | --- | --- | --- |
| data5 | 3000 | $2 \times 2$ | Super long-time, weak target |
| data8 | 399 | $2 \times 2$ | Weak target, complex background |
| data16 | 499 | $5 \times 5$ | Move fast, from far to near |
| data18 | 500 | $5 \times 5$ | Move fast, complex background |
| data19 | 1599 | $2 \times 2$ | Long-time, complex background |
| data20 | 400 | $2 \times 2$ | Weak target, target rotation |

*4.1. Qualitative Analysis*

Figure 1 shows the dynamic process of target tracking in the super long-time sequence (data5), which is used to verify the ability of the trackers to track dim targets affected by noise for a long period of time. As shown in Figure 1a, KCFD and PF deviate from the target due to the weak target, in frame 127. Additionally, we can observe in Figure 1b that methods PF, SFA-PF-TBD, and KCFD have completely lost the target before reaching frame 700, while our proposed method can better estimate the target state and stably track the target for a longer period of time, as shown in Figure 1c.

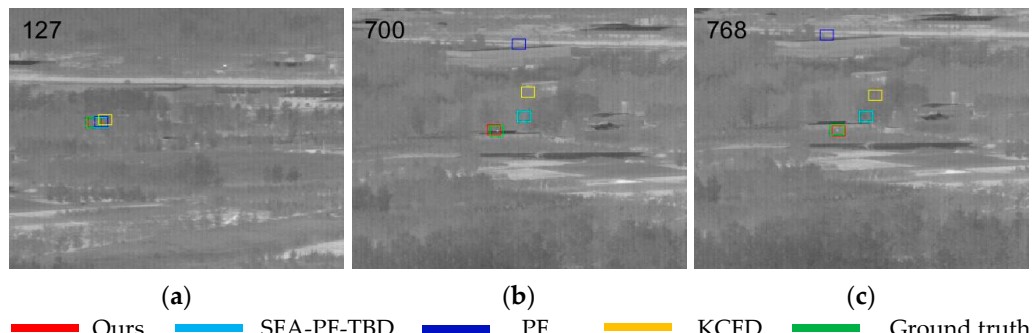

(**a**)                    (**b**)                    (**c**)

▬ Ours     ▬ SFA-PF-TBD     ▬ PF     ▬ KCFD     ▬ Ground truth

**Figure 1.** Tracking results of four methods for data5. (**a**) Results at frame 127; (**b**) results at frame 700; (**c**) results at frame 768.

Figure 2 shows the dynamic process of target tracking in the complex background sequence (data8). The images in this sequence have a complex background accompanied by objects with a grayscale similar to the target. As shown in Figure 2a, the target is obvious enough against a simple background so that all four methods can stably track the target before reaching frame 54. As shown in Figure 2b, in frame 203, objects similar to the target appear in the background, causing the KCFD method to fail in tracking the target. Similarly, in frame 366, the complicated background causes SFA-PF-TBD tracking to deviate from the target, as shown in Figure 2c. Our proposed method and PF can adapt to the complex backgrounds and robustly track the target.

Figure 3 shows the dynamic process of target tracking in the fast-moving sequence (data16). The target locations change rapidly at certain frames in data16, which requires the trackers to have the ability to rapidly retrieve the target after the target is lost. As shown in Figure 3a, the target moves suddenly and rapidly in frame 134, which causes all the methods to fail to track the target. Our proposed method and SFA-PF-TBD can rapidly retrieve the target at frame 140 and keep tracking the target successfully; PF and KCFD cannot identify and track the target, as shown in Figure 3b,c.

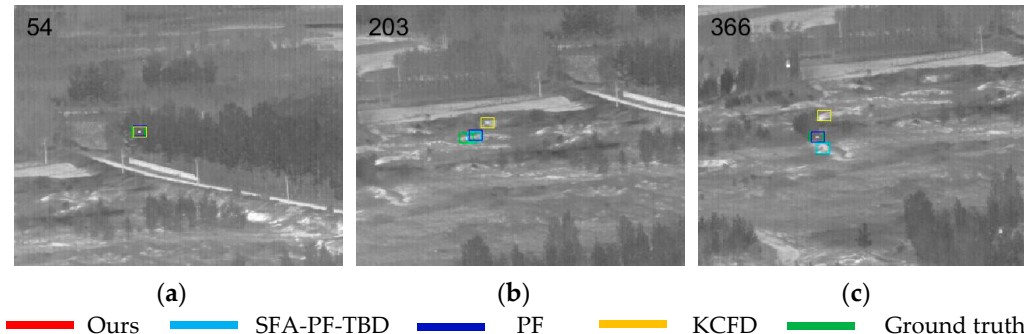

|  |  |  |  |  |
|---|---|---|---|---|
| ▬ Ours | ▬ SFA-PF-TBD | ▬ PF | ▬ KCFD | ▬ Ground truth |

**Figure 2.** Tracking results of four methods for data8. (**a**) Results at frame 54; (**b**) results at frame 203; (**c**) results at frame 366.

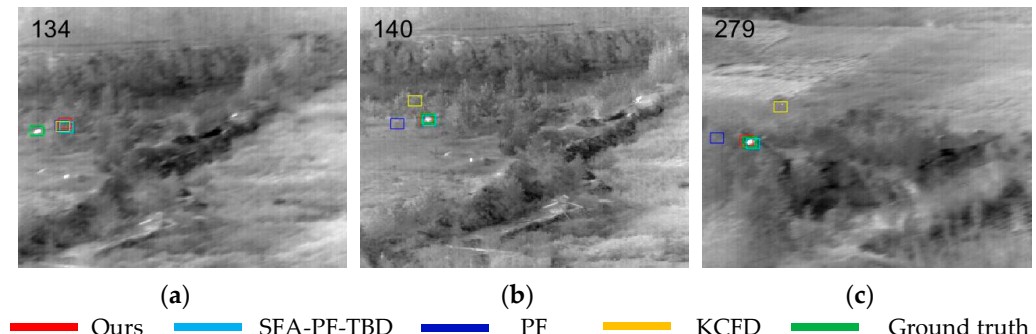

|  |  |  |  |  |
|---|---|---|---|---|
| ▬ Ours | ▬ SFA-PF-TBD | ▬ PF | ▬ KCFD | ▬ Ground truth |

**Figure 3.** Tracking results of four methods for data16. (**a**) Results at frame 134; (**b**) results at frame 140; (**c**) results at frame 279.

Figure 4 shows the dynamic process of target tracking in the fast-moving and complex background sequence (data18). As shown in Figure 4a, rapid changes in the target's position due to camera micro-vibrations and the complex background in frame 22 cause the four methods to deviate from the target. However, the proposed method and PF immediately retrieved the target at frame 27, as shown in Figure 4b, and it can be observed in Figure 4c that a deviation is produced by PF in frame 169 due to the high similarity between the target and background, while the proposed method can effectively track fast-moving targets with complex backgrounds.

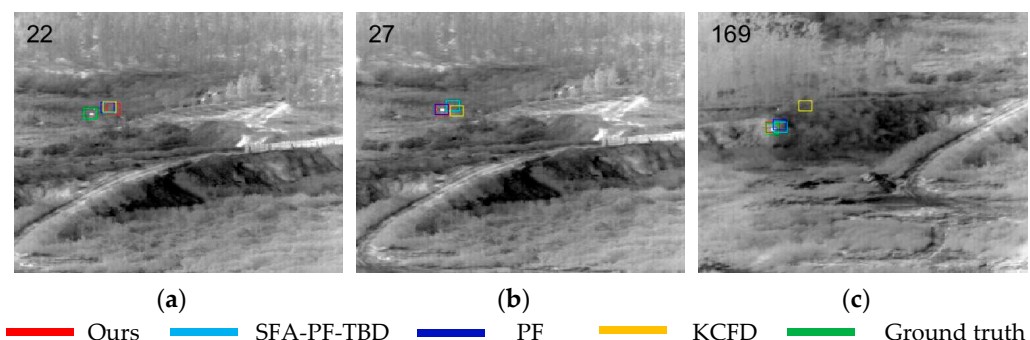

|  |  |  |  |  |
|---|---|---|---|---|
| ▬ Ours | ▬ SFA-PF-TBD | ▬ PF | ▬ KCFD | ▬ Ground truth |

**Figure 4.** Tracking results of four methods for data18. (**a**) Results at frame 22; (**b**) results at frame 27; (**c**) results at frame 169.

Figure 5 shows the dynamic process of target tracking in the long-time and complex background sequence (data19). As shown in Figure 5a,b, PF and KCFD deviate from the target and fail in their tracking processes because of the complicated background in frame 538 and frame 751. However, our method and SFA-PF-TBD can effectively track the target for a long period of time. At frame 800, SFA-PF-TBD tracking fails because the target

becomes weak, while our method only has a slight deviation and can still continue to track the target, as shown in Figure 5c.

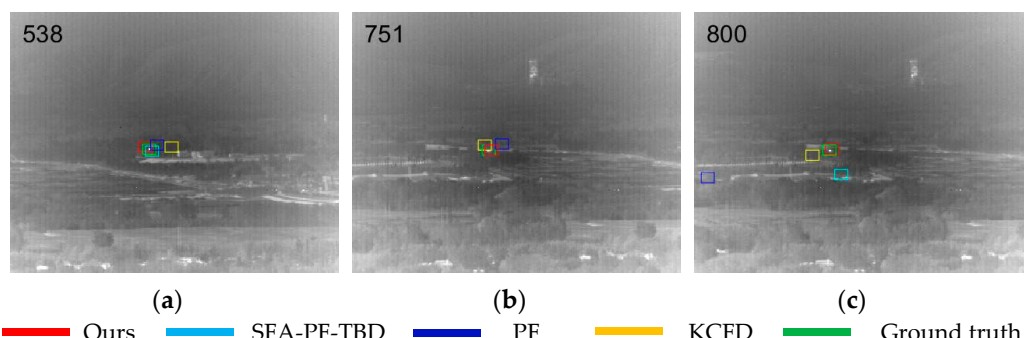

(a)        (b)        (c)

━━ Ours    ━━ SFA-PF-TBD    ━━ PF    ━━ KCFD    ━━ Ground truth

**Figure 5.** Tracking results of four methods for data19. (**a**) Results at frame 538; (**b**) results at frame 751; (**c**) results at frame 800.

Figure 6 shows the dynamic process of target tracking in the background change and target rotation sequence (data20). SFA-PF-TBD tracking failed due to the background change (from sky to mountain) in frame 238, as shown in Figure 6a. In frame 279, we can observe in Figure 6b,c that the target becomes faint due to the rotation, causing our method, SFA-PF-TBD, and PF to deviate from the target. However, when the target is recovered at frame 333, our method can continue to track the target, while others lose the target completely.

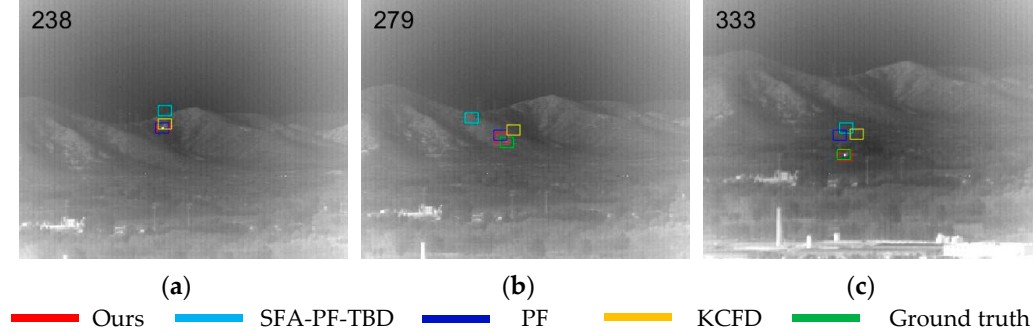

(a)        (b)        (c)

━━ Ours    ━━ SFA-PF-TBD    ━━ PF    ━━ KCFD    ━━ Ground truth

**Figure 6.** Tracking results of four methods for data20. (**a**) Results at frame 238; (**b**) results at frame 279; (**c**) results at frame 333.

*4.2. Quantitative Analysis*

To clearly compare the tracking performance of the proposed algorithm and the other three algorithms, we composed the average distance accuracy plots and average success rate plots for six groups of sequences, as shown in Figure 7a,b, respectively. It can be observed from the figure that the algorithm presented in the present paper is superior to the other three algorithms in terms of the average distance accuracy and average success rate.

Table 2 lists the average center pixel error, average 20-pixel distance accuracy, average overlap rate, and average tracking speed for six groups of sequences for the four algorithms. By comparison, it can be observed that, in terms of the average center pixel error, the proposed algorithm has the lowest error, followed by the SFA-PF-TBD algorithm. For the average 20-pixel distance accuracy, the algorithm presented in the current paper can reach up to 77.2%, and the SFA-PF-TBD is 68.6%. In relation to the average overlap rate, because the target size is too small, the overall overlap rate is low. Among the four algorithms, the average overlap rate of the proposed algorithm is the best, which is 27%. We compared the computational cost by calculating the average running speed (time to process each image) of the algorithms on the same laptop. The PF algorithm has the highest tracking speed, which can reach 121 fps, meaning it has the minimum computational cost. Compared

with SFA-PF-TBD and KCFD, the algorithm presented in the current paper has a lower computational cost that achieves a running speed of 106 fps. Therefore, on the whole, the algorithm proposed in this paper has a greater tracking performance.

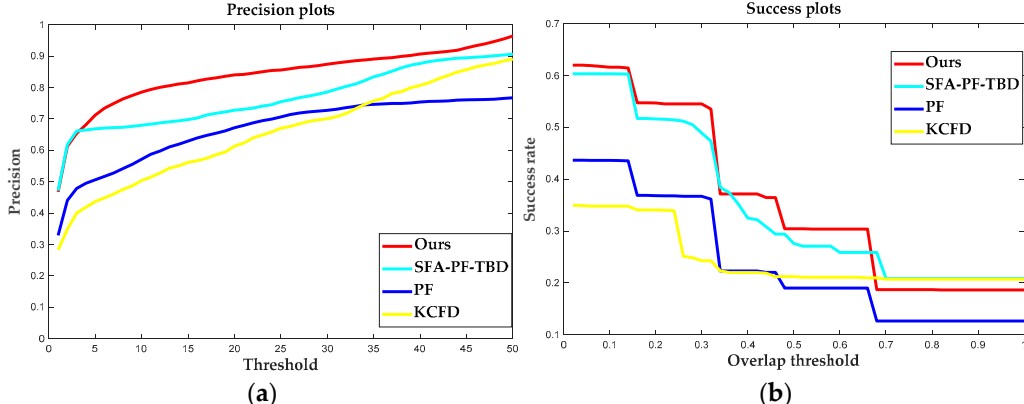

**Figure 7.** Average distance accuracy plot and success rate plot of four algorithms in six groups of sequences. (**a**) Precision plots and (**b**) success plots.

**Table 2.** Average center pixel error, 20-pixel distance accuracy, overlap rate, and running speeds of the four algorithms.

| Algorithm | Center Pixel Error/Pixels | Distance Accuracy (20) | Overlap Rate | Speed/fps |
|---|---|---|---|---|
| Ours | 37.4 | 77.2% | 27.0% | 106 |
| SFA-PF-TBD | 38.5 | 68.6% | 26.2% | 65 |
| PF | 42.9 | 66.2% | 15.9% | 121 |
| KCFD | 39.2 | 48.7% | 18.7% | 73 |

*4.3. Merits and Limitations*

The proposed method achieves stable performances for dim and small target tracking on the data set with a sky background. The fusion feature method we proposed in the present study can express dim and small targets well, which make the distance accuracy and overlap rate of the proposed method superior to the others. Moreover, the proposed method achieves a good balance between the tracking performances and computational cost, when compared with the other methods.

However, the proposed method also has some limitations. First, for some of the scenes with more complex backgrounds, a louder noise, and considerable similarities between the objects and backgrounds, our method may not accurately estimate the target state, resulting in the tracking method failing. Second, the proposed method cannot process the long-term occlusion and disappearance problems well.

**5. Conclusions**

In the current paper, we propose an improved resampling particle filter algorithm based on feature fusion, which solves the two problems in particle filters for dim and small target tracking, namely feature expression of dim and small targets and lack of the particle diversity. The fused feature of grayscale, edge, and wavelet clearly express the dim small target in various complex scenes. The improved resampling particle filter solves the problem of lacking particle diversity in a particle filter based on residual resampling with some new meaningful particles introduced, which boosts the performance of a particle filter tracker while reducing the computational complexity. The experimental results demonstrate that our method achieves a strong tracking performance with a distance accuracy of 77.2% and an overlap rate of 27%, which can effectively and robustly track dim

and small targets. Moreover, our proposed method achieves a frame rate of 106 fps with a good balance between the tracking performance and computational cost.

Although the proposed method can achieve a reasonable performance for six image sequences with sky backgrounds, it cannot track the target well in some of the more complex scenes, such as the images presenting a louder noise and long-term occlusions. The focus of future research should be to choose more reasonable features concerning adaptive fusion to improve the tracking performance for more complex scenes.

**Author Contributions:** Conceptualization, H.Z. (Haifeng Zhang) and H.W.; funding acquisition, H.W.; investigation, H.Z. (Hongbo Zhang); methodology, Y.H. and H.Z. (Hongbo Zhang); software, Y.H. and Y.C.; writing—original draft, Y.H.; writing—review and editing, Y.C. and H.Z. (Haifeng Zhang). All authors have read and agreed to the published version of the manuscript.

**Funding:** This research was funded by the West Light Foundation of the Chinese Academy of Sciences, grant No. XAB2021YN15.

**Institutional Review Board Statement:** Not applicable.

**Informed Consent Statement:** Not applicable.

**Data Availability Statement:** The data underlying the results presented in this paper are not publicly available at this time but may be obtained from the authors upon reasonable request.

**Conflicts of Interest:** The authors declare no conflict of interest.

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
