# Peer review of "Dim and Small Target Tracking Using an Improved Particle Filter Based on Adaptive Feature Fusion"

_electronics, doi:10.3390/electronics11152457_

Round 1
Reviewer 1 Report
1. Abstract
Rewrite the abstract. What is the justification for the research? What is the objective? What methodology was used? What were the main results? What were the main findings and scientific contributions?
2. Introduction
Improve the introduction. What is the importance of the topic? What are the existing works?
Related works section is missing, we recommend the author to discuss recent year papers along with the problem statement to support this research.
In contribution point 1: it's not a contribution (don’t discuss the results you achieved in this section). Mention what is supported to achieve these results.
The abbreviation should be explained clearly. Example- KCF, ECO, 64 and STRCF not discussed in first appearance.
Avoid repetition of sentences (Abstract to introduction).
What is the justification for the research? Highlight this portion.
What is the hypothesis of the research? Highlight this portion.
3. Material and Methods
Line 117 should be an equation, mention equation number.
How many features are calculated? What is the size of the feature?
F edge is not discussed well. Recheck this.
The author extracted 3 features as per the given statement. But, Equation 17 processed only f1 and f2 only. Explanation is required.
What is the impact of section 3.3? How does it improve the efficiency? Brief it. Equation 21 and 22 is pointless, explanation is required.
What was the design of the experimental test?
What statistical methods were used to analyze the results?
Include pseudo code, not its algorithm format.
4. Discussion
Result section presented very well. Results should be compared with recent related work. State of art / benchmark comparison is required to justify the effectiveness of the proposed model.
The author highlighted the computational complexity as a contribution in the introduction section. But the discussion is missing.
Additionally, the ground truth of the research should be included in the manuscript.
5. Conclusion
What are the scientific contributions? How it is solved, a summary is required.
Was the research hypothesis validated? Mention the limitation of the proposed model.
Reference section weak, we are unable to find any recent year papers. We strongly recommend the author include papers (2020, 2021 and 2022).
Reviewer 2 Report
Paper deals with important task. The authors improved resampling particle filter algorithm based on adaptive multi-feature fusion.
Paper has elements of scientific novelty and great practical value.
It has a logical structure. The paper is technically sound. The experimental section is good.
The proposed approach is logical, results are clear.
Suggestions:
1. The introduction section should be extended using more clearly the motivation of this paper.
2. The authors should add a strong related works section as there are a lot of methods in this field. It would be good to use these papers: DOI 10.1109/DSMP.2016.7583531; DOI 10.1109/CADSM.2015.7230806 among others
3. It is unclear why the authors introduced a kernel function but not used existing ones. It should be argued.
4. It would be good to explain how the proposed approach will work with the video of high resolution.
5. The conclusion section should be extended using: 1) numerical results obtained in the paper; 2) limitations of the proposed approach; 3) prospects for future research.
Round 2
Reviewer 1 Report
Thanks for the revisions.
Author Response
Thank you for your comments and suggestions.
Reviewer 2 Report
Point 2 and point 3 are not adreesed in the revision paper.
Please check the paper and fix it
